# Co-Delivery Effect of CD24 on the Immunogenicity and Lethal Challenge Protection of a DNA Vector Expressing Nucleocapsid Protein of Crimean Congo Hemorrhagic Fever Virus

**DOI:** 10.3390/v11010075

**Published:** 2019-01-17

**Authors:** Touraj Aligholipour Farzani, Alireza Hanifehnezhad, Katalin Földes, Koray Ergünay, Erkan Yilmaz, Hiba Hashim Mohamed Ali, Aykut Ozkul

**Affiliations:** 1Virology Department, Faculty of Veterinary Medicine, Ankara University, Ankara 06110, Turkey; touraj.farzani@gmail.com (T.A.F.); alireza.hanifehnezhad@gmail.com (A.H.); fldeskatalin@gmail.com (K.F.); hibaziada@yahoo.com (H.H.M.A.); 2Virology Unit, Department of Medical Microbiology, Faculty of Medicine, Hacettepe University, Ankara 06100, Turkey; ekoray@hacettepe.edu.tr; 3Biotechnology Institute, Ankara University, Ankara 06560, Turkey; eyilmaz@ankara.edu.tr

**Keywords:** CCHFV, CD24, Cytokines, nucleocapsid, IFNAR^−/−^ mice, Lethal Challenge

## Abstract

Crimean Congo hemorrhagic fever virus (CCHFV) is the causative agent of a globally-spread tick-borne zoonotic infection, with an eminent risk of fatal human disease. The imminent public health threat posed by the disseminated virus activity and lack of an approved therapeutic make CCHFV an urgent target for vaccine development. We described the construction of a DNA vector expressing a nucleocapsid protein (N) of CCHFV (pV-N13), and investigated its potential to stimulate the cytokine and total/specific antibody responses in BALB/c and a challenge experiment in IFNAR^−/−^ mice. Because of a lack of sufficient antibody stimulation towards the N protein, we have selected cluster of differentiation 24 (CD24) protein as a potential adjuvant, which has a proliferative effect on B and T cells. Overall, our N expressing construct, when administered solely or in combination with the pCD24 vector, elicited significant cellular and humoral responses in BALB/c, despite variations in the particular cytokines and total antibodies. However, the stimulated antibodies produced as a result of the N protein expression have shown no neutralizing ability in the virus neutralization assay. Furthermore, the challenge experiments revealed the protection potential of the N expressing construct in an IFNAR ^−/−^ mice model. The cytokine analysis in the IFNAR^−/−^ mice showed an elevation in the IL-6 and TNF-alpha levels. In conclusion, we have shown that targeting the S segment of CCHFV can be considered for a practical way to develop a vaccine against this virus, because of its ability to induce an immune response, which leads to protection in the challenge assays in the interferon (IFN)-gamma defective mice models. Moreover, CD24 has a prominent immunologic effect when it co-delivers with a suitable foreign gene expressing vector.

## 1. Introduction

Crimean Congo Hemorrhagic Fever (CCHF) is a tick-borne viral infection with a risk of fatal hemorrhagic disease in endemic areas. Symptomatic CCHF in humans presents as a febrile disease with severe hemorrhagic manifestations and a mortality rate of up to 40%, depending on various viral and host factors [1]. The etiological agent is Crimean Congo Hemorrhagic Fever Virus (CCHFV), classified in the Orthonairovirus genus of the Nairoviridae family. Similar to the other members of the *Nairoviridae* family, the viral genome consists of three negative sense single-stranded RNAs, frequently referred as small (S), medium (M), and large (L) segments [2].

As a highly contagious and potentially-lethal infection that is difficult to treat, prevent, and control, with the potential to cause nosocomial spread, CCHFV is an eminent target for vaccine development [3]. In general, envelope glycoproteins are the initial and most obvious viral structure considered for novel vaccine design, because of their abundance in the virion and direct relation with the host immunity. However, these regions demonstrate a substantial sequence diversity among the different CCHFV strains, especially during propagation in ticks [4]. Therefore, alternate immunization targets must also be evaluated for a broad coverage of geographically-segregated strains [5].

The nucleocapsid protein (N) of CCHFV is the main Open Reading Frame (ORF) found in the S segment and is involved in virus encapsidation. This protein is well-conserved among CCHFV strains and constitutes the immunodominant antigen with several T-cell epitopes and no known B cell targeting regions, and is considered a potential target in vaccine development for some Bunyaviruses such as the Rift Valley virus [6]. The lack of B cell epitopes due to the internal structure of this protein results in lower levels of neutralizing antibodies. However, it has been demonstrated that there is no clear association with neutralizing the antibody response and survival rates in the challenged IFNAR^−/−^ mice. It seems that balanced Th1 and Th2 responses are essential for protection in CCHFV induced disease [7].

Cluster of differentiation 24 (CD24) is a highly glycosylated mucin-like cell surface protein that presents in B and T lymphocytes, neutrophil, and macrophages. Activated B cells possessing CD24 can effectively co-stimulate the clonal expansion of CD4+ T lymphocytes [8]. As previously demonstrated, CD4+ T cells have a critical role in combating viral infections directly, as well as their enhanced effect on CD8+ T cells to engage with dendritic cells [9]. The most important known function of CD4+ T cells is to promote the high affinity neutralizing antibodies production by B lymphocyte cells [10]. Recently, evidence of these cells having a direct effect on the viral agents by the stimulation of the antiviral cytokine production or cytotoxicity effect have been identified. Naive CD24 T cells can recognize viral pathogens through antigen presenting cells (APCs). Subsequently, Th1 cells are generated because of type I interferon (IFN) and IL12 [11]. However, the presence of CD24 in the dendritic cells has the potential to down regulate T cell proliferation [12].

In this research, we assessed the immunogenicity and protection potential of the N expressing DNA vector (pV-N13) in BALB/c and IFNAR^−/−^ mice, by measuring the cytokine and total/N specific antibody responses in combination with the independently expressed CD24 (pCD24), as a potential genetic adjuvant. Based on this strategy, along with our immunization regime, we hypothesized that the simultaneous expression of CD24 may contribute to the regulation of B and T cells’ proliferation to elicit a stronger immune response in the immunized animals, leading to faster recovery.

## 2. Material and Methods

### 2.1. Ethics Statement

All of the animal experiments were performed with official permission from the Ankara University Ethical Committee for Animal Experiments (17 December 2014, 2014-23-155, and 17 October 2018, 2018-20-130) in in high containment animal facility (Animal Biosafety Level 3 plus-ABSL3+) of the department. The animal samplings were conducted according to the national regulations on the operation and procedure of animal experiments’ ethics committees (regulation no. 26220, 9 September 2006). The mice were humanely euthanized by CO_2_ exposure and cervical dislocation. The criteria for the euthanasia of the BALB/c mice was the time point of the final injection (two weeks after booster dose), and for the IFNAR^−/−^ mice, it was the positive control death time (seven days after death) and clinical symptoms (appearance changes and ocular and/or nasal discharges). Multiple observations per day were conducted so as to confirm the animals’ welfare, with constant access to autoclaved water and food provided to each individual. All of the animals, including the BALB/c and IFNAR^−/−^ mice, were purchased from B&K UNIVERSAL Ltd, Marshall Bioresouces, Hull, East Yorkshire, UK.

### 2.2. Cells

Scott and White No. 13 (SW-13) cells were used to propagate CCHFV and the perform virus neutralization assay. For indirect immunofluorescence assay (IIFA) using the immunized mice serum samples, we employed Baby Hamster Kidney (BHK-21-C13) and BHK-N (stably expressing CCHFV N) cells. The cell lines were obtained from the department’s cell culture collection and were cultured in Eagle’s Minimum Essential Medium (BHK21-C13 and BHK-N cells) and Leibovitz’s L-15 (SW-13 cells) media, supplemented with 10% heat inactivated fetal bovine serum (FBS; Biological Industries, Kibbutz Beit-Haemek, Israel), 1% penicillin-streptomycin (Biological Industries, Israel), and 1% L-glutamine (Biological Industries, Kibbutz Beit-Haemek, Israel). The individual spleens from the immunized BALB/c mice were aseptically collected and the cells were dissociated using a cell strainer. The red blood cells were then lysed using an RBC lysis buffer (Biological Industries, Kibbutz Beit-Haemek, Israel), the splenocytes were washed with 1x phosphate buffered saline (1× PBS), and the final cell pellet was suspended in RPMI 1640 media (Sigma, St. Louis, MO, USA).

### 2.3. Virus Propagation

All of the procedures related to infectious Crimean Congo Hemorrhagic Fever Virus (CCHFV) were performed in the Biosafety Level 3 plus (BSL3+) facility of the Virology Department, Faculty of Veterinary Medicine, Ankara University, Turkey. The CCHFV Ank-2 strain (access number: MK309333) was isolated from the blood sample of a patient who suffered from hemorrhagic fever in Kastamonu, Turkey. The virus isolation was performed in SW-13 cells and was verified by PCR (targeting full S and M segments), real-time PCR (targeting S segment), and sequencing (full S segment). For the virus cultivation, 90% confluent SW-13 cells in T75 culture flasks were inoculated with an Ank-2 strain at moi of 0.1 and were incubated at 37 °C up to five days so as to complete the cytopathic effects, observed as cell detachment and bubbling. The viruses were subsequently harvested, tittered, and stored at −80 °C.

### 2.4. Plasmid Construction

The RNA purification and cDNA synthesis from the CCHFV ANK-2 infected SW-13 cell supernatants were undertaken using an RNeasy Mini Kit (QIAGEN, Germantown, MD, USA) and Superscript IV First Strand Synthesis kit (Thermo Fisher Scientific, Waltham, MA, USA), according to the manufacturers’ instructions. The complete CCHFV N-coding region was amplified using Phusion High Fidelity DNA polymerase (Thermo Fisher Scientific, Waltham, MA, USA). The primers used to amplify the CCHFV N-coding region were designed based on the sequence of another local isolate (Kelkit strain, Gene Bank Accession Number: GQ337053). PCR reactions were performed for 35 cycles, as follows: initial denaturation at 98 °C for 30 s, denaturation at 98 °C for 10 s, annealing at 55 °C for 30 s, and extension at 72 °C for 40 s followed by a final extension of 10 min at 72 °C. The PCR products were gel purified and cloned into pVAX-1 (Invitrogen, Carlsbad, CA, USA) using Seamless Ligation Cloning Extract (SLiCE), which was prepared from *Escherichia coli* strain PPY (a DH10B-derived strain containing an optimized λ prophage Red recombination system), as described previously, to construct pV-N13 [13]. The PCR product used in the SLiCE reaction was flanked by 50 bp homologous arms to the EcoRI site of the pVAX-1 vector’s MCS, in order to guarantee in vitro homologous recombination (Table 1). The CD24 amplification (Table 1) and subsequent pCD24 construction were performed based on the cloning of CD24 in pVAX-1, as described previously [14]. The pCD-N1 plasmid was also developed by the insertion of an S segment in a pCDNA3.1 myc/HisA vector (Thermo Fisher Scientific, Waltham, MA, USA), by digestion of the PCR product and vector with FastDigest EcoRI and XhoI (Thermo Fisher Scientific, Waltham, MA, USA) restriction enzymes (Table 1), followed by ligation with T4 DNA ligase enzyme (Thermo Fisher Scientific, Waltham, MA, USA). Colony PCR, restriction enzyme digestion, and next generation sequencing verified all of the constructs. Endotoxin-free pV-N13 and pCD24 plasmids were prepared by Purelink Expi Endotoxin-free Maxiprep Kit (Thermo Fisher Scientific, Waltham, MA, USA) and were quantitated (NanoDrop™ One/OneC Microvolume UV-Vis Spectrophotometer, Thermo Fisher Scientific, Waltham, MA, USA) prior to immunization.

### 2.5. In Vitro Protein Expression

In order to conduct an indirect immunofluorescence assay (IIFA), following 24 h of cell cultivation in 24 well plate, the pV-N13 and pCD24 plasmids were separately transfected into BHK21-C13 cells using Lipofectamine 3000 (Thermo Fisher Scientific, Waltham, MA, USA), according to the manufacturer’s instruction. Forty-eight hours post-transfection, the cells were fixed with 3.7% formaldehyde, followed by blocking with 5% skimmed milk (Cell Signaling, Leiden, The Netherlands) in 1× Tris-buffered saline (TBS) buffer containing 0.2% Tween-20 (1× TBST) for 90 minutes. For the pV-N13 construct, anti-CCHFV-N human polyclonal antibody, and for the pCD24 vector, the primary antibody of the CD24 FITC-antibody (Thermo Fisher Scientific, Waltham, MA, USA) were diluted by 1:250 in a 1× TBST buffer, then added to each well, followed by overnight incubation at +4 °C. Incubation with a secondary antibody (FITC-labelled anti-Mouse and anti-human Whole IgG; Sigma, St. Louis, MO, USA) was performed at room temperature for 1 h. The immunoreactive cells were visualized by examination with an Axio Vert A1 Microscope (Ziess, Oberkochen, Germany).

For the Western blot assay, pV-N13 transfected BHK21-C13 cells were scraped 72 h post-DNA delivery, and were collected and lyzed using PRO-PREP™ Protein Extraction Solution (iNtRON Biotechnology, Burlington, MA, USA), according to the manufacturer’s instruction. After quantitation using a Bradford assay kit (Thermo Fisher Scientific, Waltham, MA, USA), the proteins (30 µg/well) were separated by Mini-Protean TGX Stain free precast gels (BioRad, Inc., Hercules, CA, USA) in a 1x Tris/Glycine/SDS buffer (BioRad, Hercules, CA, USA), and were transferred to the PVD membrane using the Trans-Blot Turbo Transfer System (BioRad, Hercules, CA, USA). After the immobilization of the bands, blocking and incubation with the primary antibody were performed, as described earlier. The membrane was treated with anti-human-IgG-HRP conjugate (Sigma, St. Louis, MO, USA) at room temperature for one hour. The bands were visualized after incubation with Clarity Western enhanced chemiluminescence (ECL) substrate solution (BioRad, Hercules, CA, USA) for 10 minutes in the dark, and were imaged using the ChemiDoc MP System (BioRad, Hercules, CA, USA).

### 2.6. BHK-N Cells

To perform IIFA from the serum samples of the immunized mice, we developed a BHK21-C13 cell line that is stably transfected by a pCD-N1 plasmid. For this purpose, after transfection by Lipofectamine 3000 (Thermo Fisher Scientific, Waltham, MA, USA), the cells were further incubated in the presence of 500 µg/µL geneticin (G418) and 10% FBS for two weeks. The survived colonies were collected and pooled and continued to grow in the Dulbecco’s Modified Eagle’s medium (DMEM) containing the selective antibiotic pressure, which was 500 µg/µL of G418. The stable cells were split 1:3 each week, and before IIFA, the expression of N protein was verified by immunostaining, using a mice monoclonal antibody targeting N protein as a positive control, which gave a positive rate of 80%–90% by examining the green fluorescence.

### 2.7. Immunization of BALB/c Mice

A total of twenty 8–10-week-old female BALB/c mice were randomly divided into five groups, including pV-N13, pCD24, pV-N13 plus pCD24, pVAX-1, and normal saline. All of the groups, except for the normal saline, received 50 µg of DNA constructs in a volume of 300 µL. For the pV-N13 plus pCD24 group, we previously tried different regimes ranging from 1:1 (µg) to 5:1 (µg), and the best results based on cytokine and total/specific antibody responses were obtained at 4:1 ratio (pV-N13 to pCD24). The immunization was repeated twice through the intramuscular route (thigh muscles), with an interval of two weeks. Blood samples were collected from the tail vein based on our laboratory’s routine, on days 0, 14, and 28, and the serums were used in neutralization, IIFA, cytokine, and total/specific antibody assays. In addition, the immunized mice were humanely euthanized on day 28 and the splenocytes from each individual were collected for in vitro cytokine assay (Figure 1A,B).

### 2.8. Challenge Experiments

Prior to the mice challenge, the lethal doses of four different local CCHFV strains isolated from human cases were detected in 8–10-week-old female IFNAR^−/−^ mice. These strains (Ank-2, Ank-15, Ank-16, and Ank-58) were isolated from the blood samples of CCHFV infected humans by propagation in SW-13 cells, and were verified as CCHFV by PCR and next generation sequencing of S segment, and two of them were submitted to GenBank (Ank-2: MK309333 and Ank-58: MK309334). The IFNAR^−/−^ mice were inoculated intraperitoneally with 300 µL of virus dilutions (10, 100, 1000, and 10,000 Median Tissue Culture Infectious Dose; TCID_50_) in triplicate and were observed twice a day for symptoms, including appearance changes, depression, weight loss, and death. Following the detection of the suitable strain and relevant lethal dose, a total of 20 IFNAR^−/−^ mice were immunized based on the given schedule for the BALB/c mice, and the animals were challenged with 300 µL of 100 Median Lethal Dose; LD_50_ (1000 TCID_50_) of Ank-2 strain (third passage in SW-13 cells) two weeks after the final booster. The inoculated mice were observed daily for symptom development during the next 13 days’ post challenge. The cytokine responses in the challenged mice were analyzed using the serum samples collected on days 0 (before immunization with vaccine constructs), 28 (2 weeks after booster dose and before challenge), and from the animals at the end of their experiments (on the day of euthanasia for the control groups or day 13 for the surviving immunization groups).

### 2.9. Quantitative (q) PCR

The total RNAs from the challenged IFNAR^−/−^ mice organs (brain, spleen, and liver) were extracted using TRI-REAGENT (BioShop Inc, Burlington, Ontario, Canada), in accordance with the manufacturer’s instruction. Briefly, the tissue samples were homogenized in a TRI Reagent (1 mL/50–100 mg tissue), and then stored for 5 min at room temperature. Next, 0.2 mL of chloroform was added, and mixed vigorously for 15 s. The resulting mixture was stored at room temperature for 15 min and centrifuged at 12,000 *g* for 15 min at 4 °C. The aqueous phase was transferred to a fresh tube and the RNAs were precipitated by mixing with 0.5 mL of isopropanol. The samples were then incubated at room temperature for 10 min, followed by centrifugation at 12,000 *g* for 8 min at 4 °C. The RNA pellet was washed with 1 mL 75% ethanol and was centrifugated at 7500 *g* for 5 min at 4 °C and were dissolved in ultra-pure molecular-grade water. The quantitative PCR (qPCR) assay was carried out using the Rotor-Gene Q instrument (QIAGEN, Germantown, MD, USA) and QuantiNova^®^ Pathogen +IC Kit (QIAGEN, Germantown, MD, USA). The final master mix (15 μL) comprised of 5 μL of 4× Reaction Mix, 6.3 μL of PCR-grade water, 1.6 μL of each primer (at 0.8 μM final concentration), and 0.5 μL of probe (at 0.25 μM final concentration) (Table 1). Then, 5 μL of the template RNA (500 ng) was added to the master mix. The cycling conditions that were used were 50 °C for 10 min, 95 °C for 2 min, followed by 40 cycles of 95 °C for 5 s, and 60 °C for 30 s.

### 2.10. Indirect Immunofluorescence Assay (IIFA)

The presence of N specific antibodies (IgG) was analyzed by IIFA in stable BHK-N cells. For this purpose, heat inactivated serums from the BALB/c mice (taken on day 28) and a 1/50 dilution of each one were added to four wells of a 96-well plate containing confluent BHK-N, previously fixed by 3.7% formaldehyde and blocked using 5% skimmed milk in 1× TBST buffer. After a 90-min incubation at room temperature (RT), the secondary antibody of the FITC-labelled anti-Mouse IgG (whole molecule) was added to the wells. Following one-hour incubation at RT, the cells were visualized using a fluorescence microscope.

### 2.11. Virus Neutralization Assay (VNA)

The neutralizing activity of the anti-N antibodies in the immunized BALB/c and IFNAR^−/−^ mice were evaluated via VNA, using the CCHFV Ank-2 isolate. Briefly, the serum samples were inactivated at 56 °C for 30 min in a water bath, and were serially-diluted (two-fold in DMEM) and mixed with an equal volume of 100 TCID_50_ virus titer in duplicate, and incubated for 1 h at 37 °C. The serum–virus mixtures were subsequently inoculated onto one-day-old 90% confluent SW-13 cells and were grown in 24-well plates. The infected cells were further incubated under the identical conditions for four to five days, with daily observation via inverted microscope for the virus-induced cytopathic effects.

### 2.12. Total Antibody Isotyping

The serum samples obtained from the immunized BALB/c mice on days 0 (before first injection) and 28 (before euthanasia) were used for the determination of the total antibody isotypes, using a Total Antibody Isotyping ELISA kit (Thermo Fisher Scientific, Waltham, MA, USA). Briefly, 50 µL of diluted serums (1:500 in DMEM) was mixed with an equal volume of goat anti-mouse IgG plus IgA plus IgM HRP conjugate, and was kept at room temperature for 1 h. Following three washing steps, 75 µL of the TMB substrate was added. After 15 min, a TMB stop solution was added and the reaction was read at 450 nm in an ELISA reader (Titertek Multiskan PLUS MK II Microplate Reader, Midland, ON, Canada).

### 2.13. Cytokine Assay

The splenocyte from the immunized BALB/c mice were dispersed to 24-well plates at a concentration of 2.5 × 10^5^ cells per well, and were immediately infected with 10 moi of CCHFV Ank-2 strain. The infected cells were further incubated for 72 h at 37 °C and a 5% CO_2_ atmosphere. As controls, we included naïve and virus stimulated splenocytes from the negative controls. The culture supernatants were collected at 48 and 72 h of incubation, and were stored at −80 °C until further use. The measurement of the cytokines derived from the serum samples of the BALB/c (on days 0 and 28) and IFNAR^−/−^ mice (on days 0, 28, and 41) alongside the BALB/c splenocyte supernatants were performed using a LEGENDplex™ Mouse Th_1_/Th_2_ Cytokine Panel (8-plex) kit (BioLegend, San Diego, CA, USA), as described by the manufacturer, using a FacsCanto II Flow Cytometer platform (BD Bioscience, Franklin Lakes, NJ, USA).

### 2.14. Statistical Analysis

The antibody isotyping data and cytokine levels among the groups were evaluated using two-way (Sidak’s post hoc correction) ANOVA by SPSS (IBM SPSS Statistics for Windows, Version 22.0. Armonk, NY, USA). All of the data were presented as mean ± standard deviation (SD). Graphs were produced using GraphPad Prism version 7.0 (GraphPad Software, San Diego, CA, USA; www.graphpad.com). The statistical significance level was determined as 0.05. All of the molecular biology procedures were simulated using SnapGene Viewer software (www.snapgene.com).

## 3. Results

### 3.1. Vector Construction and In Vitro Expression of CCHFV N

After the virus (Ank-2 strain) propagation in the SW-13 cells (Figure 2B) and the PCR amplification of the S segment from the extracted RNAs with no codon optimization, the pV-N13 plasmid was constructed via the SLiCE approach, because of its simplicity and efficiency (Figure 2C). All of the colonies obtained from this approach showed the correct structure in the restriction enzyme analysis and sequencing. In addition, a pCD24 construct was generated by the standard cloning method. In IIFA, the CCHFV N (Figure 2D,E) and CD24 (Figure 2F,G) expression could be visualized as green fluorescence 48 h post transfection. Furthermore, in immunoblotting, 52 kDa N protein was detected in the cells after 72 h post-transfection of the pV-N13 construct (Figure 2H).

### 3.2. Serological Assays

During the immunization of the BALB/c mice, no signs of disease or allergy were recorded in all of the groups. The pCD24 construct induced the highest level of total antibody isotypes among the study groups, except for IgM, which was significantly elevated in the pV-N13 plus pCD24 group. The pV-N13 plus pCD24 and pV-N13 groups showed a relatively significant amount of IgG1, IgG2a, IgG2b, and IgG3 antibodies (Figure 3A–E). The IgG2a/IgG1 ratio was also determined and demonstrated to be <1 in all groups of BALB/c mice, except the saline control, which indicates a shift to the Th2 response because of the immunization schedule (Figure 3F). In IIFA, the serum samples (day 28) from the BALB/c groups of pV-N13 and pV-N13 plus pCD24 were observed to bind the stably expressed N in BHK-N cells (Figure 3G–H). Despite the high amounts of antibodies produced, all of the serum samples of both mice models subjected to VNA revealed a lack of neutralizing activity at 14- and 28-days post immunization.

### 3.3. Cytokine Assay in BALB/c and IFNAR^−/−^ Mice

The virus stimulated splenocytes from the pV-N13 and pV-N13 plus pCD24 immunized BALB/c mice showed a significant amount of IFN-gamma and IL-2 cytokines after 48- and 72-hours cultivation. These cytokine responses were comparably elevated in the pCD24 group, despite the lack of in vitro viral protein expression (Figure 4A,B). The pV-N13 plus pCD24 group also produced sufficient levels of IFN-gamma in the serum. Moreover, the pV-N13 and pV-N13 plus pCD24 groups revealed a prominent potential for IL-2 stimulation (Figure 5A,B). The pV-N13 plus pCD24 group showed the highest levels of IL-4, IL-5, and IL-13 secretion in the supernatant of the virus stimulated splenocytes. Moreover, the pV-N13 immunized mice also revealed the potential to produce a significant amount of IL-4 and IL-13. In spite of IL-13, the pCD24 group had diminished responses of IL-4 and IL-5 in the supernatants (Figure 4C–E). In the serum samples, the significantly elevated levels of IL-4, IL-5, and IL-13 were observed in the pV-N13 plus pCD24, pV-N13, and pCD24 groups. The mice receiving pV-N13 plus pCD24 demonstrated the highest IL-5 production (Figure 5C–E). In the supernatants, the highest amounts of TNF-alpha and IL-6 cytokines were detected in the pV-N13 plus pCD24 group. The pCD24 immunized mice showed significantly elevated IL-10 levels. Furthermore, the TNF-alpha and IL-6 secretion were relatively increased in the pV-N13 group (Figure 4G–H). IL-10 and TNF-alpha is pronounced in pV-N13 plus pCD24 immunized mice group’s serums. On the other hand, the IL-6 production was markedly increased in the serums of the pCD24 group (Figure 5G–H). As in vitro control groups for splenocyte supernatants, we included naïve and Ank-2 virus stimulated splenocytes from normal saline injected BALB/c mice.

In the challenged IFNAR^−/−^ mice serum samples, the IFN-gamma level in the pV-N13 plus pCD24 and pCD24 groups are elevated on day 28. In addition, the IFN-gamma is also significantly stimulated in the pV-N13 group after challenge with Ank-2 strain (Figure 6A). The pV-N13 plus pCD24, pV-N13, and pCD24 groups have further demonstrated a marked potential for IL-2 and IL-4 stimulation. Among these groups, the pV-N13 plus pCD24 is predominant in IL-2, as well as IL-5 secretion (Figure 6B,C). Moreover, the pV-N13 and pCD24 groups also have relatively increased IL-5 levels in the sera (Figure 6D). Furthermore, in the IFNAR^−/−^ mice, the pV-N13 plus pCD24 and pV-N13 groups reveal an elevation of IL-6 (Figure 6E). Also, the pV-N13 plus pCD24 and pV-N13 groups are predominant in the TNF-alpha in comparison to other mice groups (Figure 6F).

### 3.4. Challenge Studies

Three local CCHFV strains produced a mortality in the IFNAR^−/−^ mice during in vivo challenge assays (Figure 7A). The infected IFNAR^−/−^ mice with the Ank-58 strain survived the challenge experiment. To verify the virus replication in vivo, we conducted a TCID_50_ assay by using organ specimens, all of which showed high viral loads. The sequencing of the S segment showed no amino acid substitution in this strain. By considering the growth performance of the viruses in the SW-13 cells and the infectivity in the IFNAR^−/−^ mice (LD50 = 1:1000/0,3 mL), CCHFV Ank-2 strain was selected as the challenge virus. The challenge assay was performed in the immunized IFNAR^−/−^ mice two weeks following the final booster dose on day 28. The pV-N13 and pV-N13 plus pCD24 groups survived after challenge and the body weight in both groups were almost stable, except for the slight decrease during the first days (Figure 7B,C). The clinical signs of these two N expressing groups were limited to weight loss during the first days of challenge. The animals immunized with normal saline, pVAX1, and pCD24 showed symptoms including weight loss, appearance change, depression, and ocular discharge before death, on days 5, 6, and 8, respectively (Figure 7B).

### 3.5. qPCR

As expected, the pV-N13 and pV-N13 plus pCD24 groups demonstrated the lowest virus copy number/µL in the brain, spleen, and liver, 13 days post challenge by the Ank-2 strain. Among the pV-N13 and pV-N13 plus pCD24 groups, pV-N13 showed a relatively faster but non-significant clearance of virus in these organs (Figure 8A–E).

## 4. Discussion

The efforts for finding a suitable gene of CCHFV as a vaccine target are well-justified, because of the imminent public health threat posed by the disseminated virus activity in the broad geographical regions and the lack of an approved therapeutic regime. While a vaccine licensed for human use would decrease the disease incidence and infection-associated morbidity and mortality, animal vaccination is likely to have an impact on the vector and reservoir control, resulting in a reduction of zoonotic transmission [3].

The DNA vector-based delivery of immunogenic viral proteins, such as glycoproteins and nucleocapsid, is a practical approach for CCHFV vaccine development, as any other strategy requiring virus handling would necessitate high biological containment facilities [15]. Currently, the immune responses during CCHFV infection are not well understood, with scarce data on T and B cell epitopes in virally-encoded proteins [16]. Studies based on glycoproteins (precursor Gc and Gn) and nucleocapsid have revealed contradictory results for protection in challenges of the IFNAR^−/−^ and immune suppressed (IS) mice models [5]. However, it is generally accepted that the activation of the CD4^+^ and CD8^+^ responses are involved in the immune control of infection severity, and the neutralizing antibodies play little role in protection. Therefore, we focused on the S segment, suggested as an ideal target in protection assays in various platforms, such as modified Vaccinia Ankara Virus (MVA), Adenovirus (AdV), virus-like particles (VLP), and DNA vectors [5,6,7]. It is also important to assess the probable correlation between the challenge protection and CD4^+^ and CD8^+^ responses [17].

Despite the built-in adjuvant of the DNA vectors (CpG), this gene transfer platform is less immunogenic when compared to live or subunit vaccines [18]. The co-delivery of the genetic adjuvants during DNA immunization can substantially enhance the immunogenicity of the expressed antigens. In particular, cytokines (GM-CSF, IL-1, TGF-b, and IFN-gamma) and co-stimulatory factors (CD80, CD86, and CD40L) have been explored as genetic adjuvants in different settings [19,20,21,22,23,24,25,26]. Various factors, such as the administration route and co-injection of additional potentiators have been reported to affect the outcomes of genetic adjuvants. The intradermal but not intramuscular co-injection of CD80 with the herpes simplex virus antigens elicited protection from the virus challenge [27]. In another study, the co-administration of CD40L with b-galactosidase resulted in a striking increase in the antigen specific production of IFN-gamma, cytolytic T cell activity, and IgG2a antibodies, indicating a Th1 bias [26].

CD24 acts as a co-stimulatory factor of T lymphocyte homeostasis and proliferation, and is involved in B lymphocyte activation and differentiation. This protein stimulates the antigen-dependent proliferation of B lymphocytes, and prevents their terminal differentiation into antibody producing cells [28].

In this study, we evaluated an eukaryotic expression vector (pVAX-1) expressing a CCHFV nucleocapsid protein (pV-N13), in order to conduct a challenge experiment and to assess the total and specific immune responses against the S segment of CCHFV in IFNAR^−/−^ and BALB/c mice models. In addition, we investigated the potential impact of CD24 as a potential genetic adjuvant in this setting. The pVAX-1 based CD24 expression vector (pCD24) was delivered individually and/or in combination with the N expressing construct. We observed that the individual delivery of pCD24 could induce cytokine and total antibody responses, alongside increased total IgG subtypes, surpassing the N expressing constructs in the BALB/c mice. Furthermore, pCD24 demonstrated a prominent effect when administered simultaneously with the pV-N13 vector, on Th1 and Th2 cytokines induction, noted as having statistically-significant elevations of IFN-gamma, IL-2, IL-4, IL-5, and TNF-alpha in the virus stimulated splenocytes of the BALB/c mice and all of the cytokines measured in the immunized BALB/c and IFNAR^−/−^ mice serum samples. A similar finding was also noted in the total IgG1 and IgG2a production, despite a lack of statistically-significant differences in the BALB/c mice model. Therefore, an overall synergistic effect of the pCD24 and pV-N13 construct was noted in this level. Evaluation of the total immunoglobulin isotypes in the immunized BALB/c mice further confirmed the capacity of the N expressing construct to induce significant amounts of total IgG1, IgG2a, IgG2b, IgG3, and IgM antibodies. Interestingly, the CD24 expressing plasmid produced significantly higher levels of IgG subtypes when injected individually. Despite the synergistic effect of the CD24 expression on the total antibody production against pV-N13, we identified that the produced antibodies have no in vitro neutralizing activity in both of the mice models. These findings do not undermine N as a vaccine target, as it has been previously documented in animal models that non-neutralizing antibodies could protect mice model from lethal challenges [3,6]. A similar phenomenon was also described in influenza vaccine studies with animals, where protective anti-influenza immunity is attained in the absence of measurable neutralizing antibodies [29]. In CCHFV infections in STAT-1 knockout mice, a low level of neutralizing antibodies was observed, and cell-mediated immunity and interferon induction seem to play a decisive role in the outcome [17]. For a deeper investigation of this phenomenon, we conducted challenge assays in IFNAR^−/−^ mice because of their defect in IFNαβ receptors. CCHFV is considered as an IFNαβ sensitive virus, so challenge experiments were conducted in this mice breed [30]. In this experiment, a protective rate of 100% was documented in the pV-N13 and pV-N13 plus pCD24 groups. Moreover, by analyzing the cytokine responses in the IFNAR^−/−^ mice on days 0, 28 (after booster dose and before challenge), and different days after challenge (depends on the death of the control groups or on the euthanasia of the N expressing groups on day 41), it is obvious that the elevation of TNF-alpha and IL-6 plays an important role in protection against this disease. Other cytokines, including IFN-gamma, IL-2, IL-4, and IL-5, are also increased in the pCD24 group, demonstrating that they do not significantly affect disease progression in the infected IFNAR^−/−^ mice. The adjuvant potential of CD24 in our challenge experiment was not clear in the survival assay, despite the fact that its obvious role is in mitigating disease severity. On the other hand, pV-N13 plus pCD24 group showed a more stable percentage of body weight in comparison to the pV-N13 group. These data support our other findings and indicate that pV-N13 plus pCD24 has better potential to stimulate IL-6 and TNF-alpha secretion in the IFNAR^−/−^ mice model. Furthermore, defective IFN-gamma receptors, which lead to different secretion patterns of other cytokines in these mice, render data analysis more complicated, so it is not easy to interpret the challenge assay and make a general hypothesis in BALB/c mice and humans.

In conclusion, because of the importance of CCHFV in the endemic areas like Turkey, an urgent attempt to develop an efficient vaccine based on local isolates must be conducted. Moreover, the route of transmission in the ticks, which exposes the virus to mutation, makes the attempts to fight the disease more strenuous. Our findings along with others emphasize this fact that nucleocapsid protein can be considered as an attractive target in immunization because of some of its special characteristics, like preservation and the ability to stimulate balanced Th1 and Th2 responses. In addition, we have evaluated the potential of CD24 as a new genetic adjuvant in the DNA vaccination of mice models supplemented with the gene of interest, and despite the clear findings related to the immune responses (especially cytokines), it is impossible to clearly infer its potential as a new adjuvant, at least in this experiment. The effect of this protein in the immune system must be investigated more in order to ascertain any correlation among it and the different arms of innate immune responses.

## Figures and Tables

**Figure 1 viruses-11-00075-f001:**
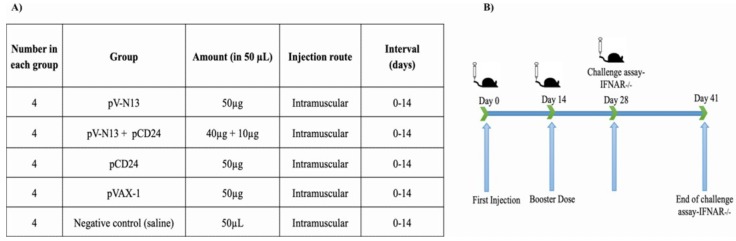
Immunization scheme.

**Figure 2 viruses-11-00075-f002:**
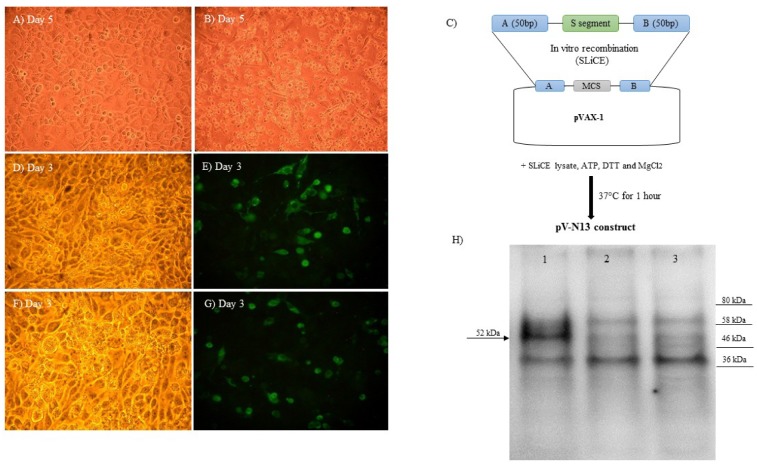
(**A**,**B**) Propagation of Ank-2 strain of Crimean Congo hemorrhagic fever virus (CCHFV) in Scott and White No. 13 (SW-13) cells on day five (A: cell control, B: virus infected cells). Magnification ×400. (**C**) Plasmid construct used for DNA immunization in this study. In vitro homologous recombination between the empty vector and amplified nucleocapsid, flanked by 50 bp homologous arms to the EcoRI recognition sequence of the vector multiple cloning site, occurs in the presence of the Seamless Ligation Cloning Extract (SLiCE) lysate from PPY bacteria, adenosine triphosphate (ATP), 1,4-Dithiothreitol (DTT), and MgCl_2_ at 37 °C. (**D**–**G**) N protein expression in the cells transfected by pV-N13 via indirect immunofluorescence assay (IIFA) 72 hours post DNA delivery (D: phase contrast; E: fluorescent contrast). CD24 protein expression in the cells transfected by pCD24 via IIFA after 72 h (F: phase contrast; G: fluorescent contrast). (**H**) Western blot analysis of the BHK21-C13 cells transfected with pV-N13. The expected protein (~52 kDa) was detected after 72 h (lane 1). We included pVAX-1 transfected cells (lane 2) and cell control (lane 3) in the assay.

**Figure 3 viruses-11-00075-f003:**
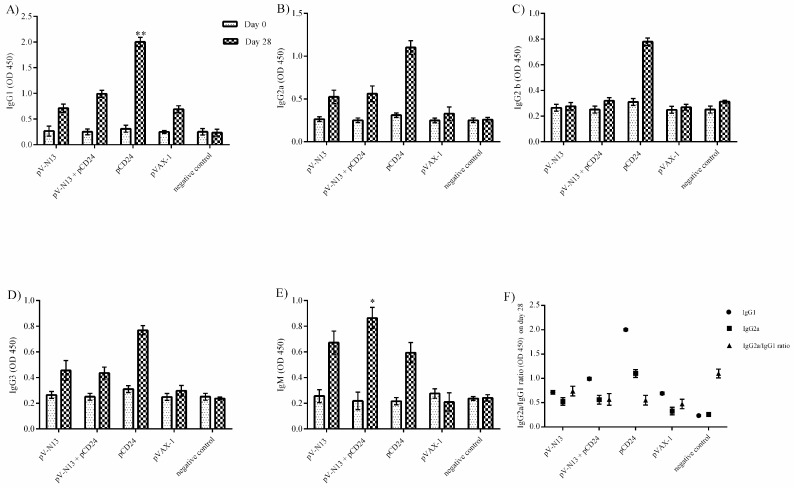
Serological assays (**A**) IgG1 response. (**B**) IgG2a response. (**C**) IgG2b response. (**D**) IgG3 response: In all of the mentioned assays, the pV-N13 plus pCD24 and pCD24 groups are dominant. (**E**) IgM response: pV-N13, pV-N13 plus pCD24, and pCD24 groups are a stimulator of the IgM response. (**F**) Comparison of IgG1, IgG2a, and IgG2a/IgG1 ratio responses. * *p* < 0.05; ** *p* < 0.01; and *** *p* < 0.001 versus the pVAX-1 group. All of the data are shown as mean ± standard deviation (SD). (**G**–**H**) Detection of the N specific antibodies present in the serum samples in the Baby Hamster Kidney (BHK)-N cells (G: pV-N13 immunized mice serum samples, H: pV-N13 plus pCD24 immunized mice serum samples).

**Figure 4 viruses-11-00075-f004:**
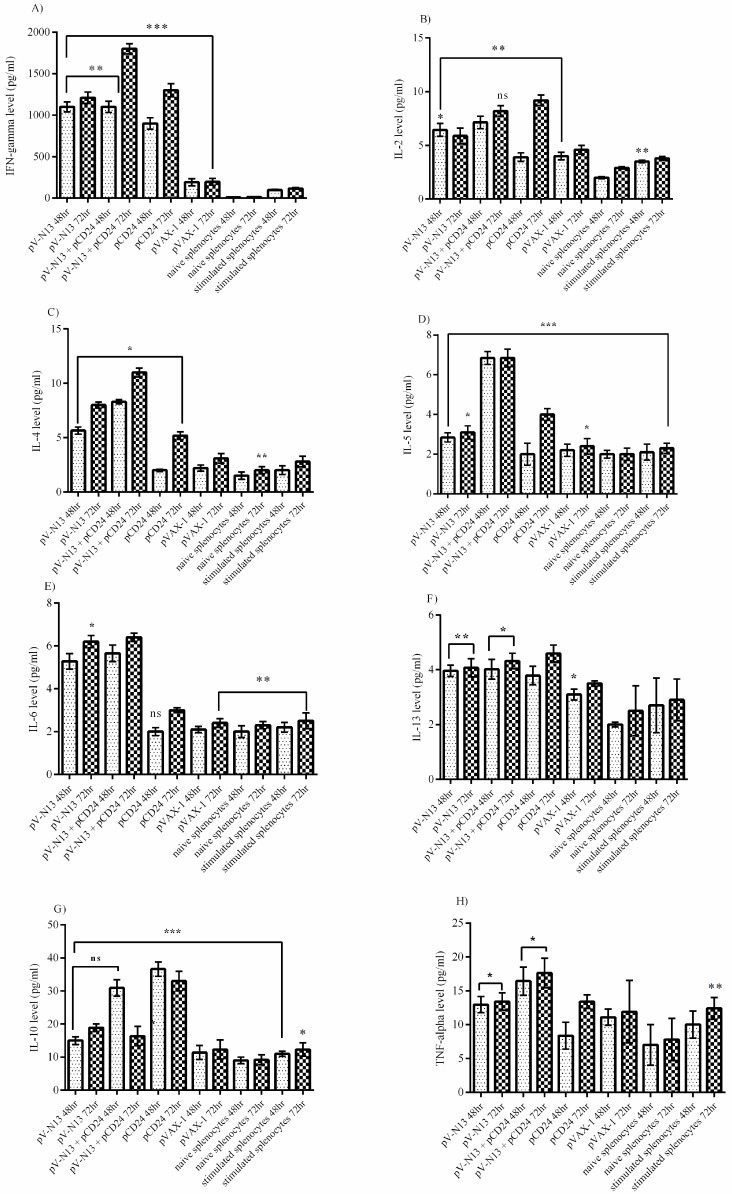
Cytokine responses in the supernatant of CCHFV stimulated splenocytes of the immunized BALB/c mice. (**A**) Interferon (IFN)-gamma response: As demonstrated here, the pV-N13 plus pCD24 group’s result is higher in comparison with the other groups. The pCD24 stimulation level is also significant. (**B**) IL-2 response: IL-2 response is predominant in the pV-N13 plus pCD24 and pCD24 groups. (**C**) IL-4 response: As shown, the pV-N13 plus pCD24 and pV-N13 immunized BALB/c mice demonstrated the highest amount. (**D**) IL-5 response: the IL-5 response is predominant in the pV-N13 plus pCD24 and pCD24 groups. (**E**) IL-6 response: pV-N13 plus pCD24 and pV-N13 groups are higher in comparison with the other groups. (**F**) IL-13 response: All of the immunized mice showed almost identical levels of IL-13 secretion. (**G**) IL-10 response: When comparing the N expressing groups to the empty vectors, the pV-N13 plus pCD24 group elicited a pronounced IL-10 response in the BALB/c mice model. Also, pCD24 has the potential to induce IL-10 responses. (**H**) TNF-alpha response: As shown, the pV-N13 plus pCD24 group stimulated the highest amount. * *p* < 0.05; ** *p* < 0.01, and *** *p* < 0.001 versus the pVAX-1 group. The ‘ns’ in the graphs indicates non-significant data. All of the data are shown as mean ± SD.

**Figure 5 viruses-11-00075-f005:**
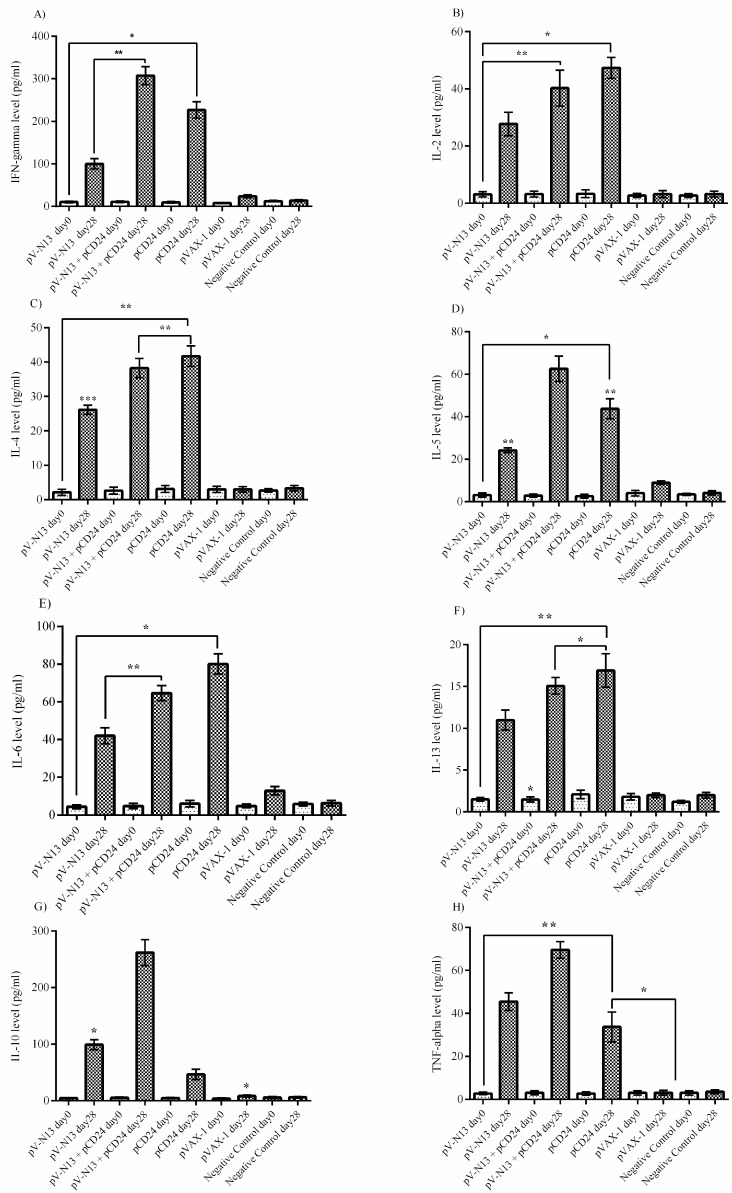
Cytokine responses in the serum samples of the immunized BALB/c mice. (**A**) IFN-gamma response: pCD24 vector (on day 28) stimulated a prominent production of IFN-gamma. pV-N13 plus pCD24 on day 28 is by far dominant when compared with other groups. (**B**) IL-2 response: pCD24 and pV-N13 plus pCD24 possessed the highest amount. Moreover, the IL-2 level in the pV-N13 group is also adequate. (**C**) IL-4 response: IL-4 stimulation in pCD24 and pV-N13 plus pCD24 is considerable when compared with the other groups. (**D**) IL-5 response: Interestingly, the pV-N13 plus pCD24 group has potential to stimulate this cytokine in immunized mice in a higher level compared to the remaining groups. The amount of IL-5 in the pCD24 immunized mice is also significantly elevated. (**E**) IL-6 response: pCD24 and pV-N13 plus pCD24 immunized mice possessed the highest level of IL-6 in the serum samples. (**F**) IL-10 response: The results indicated the predominance of the pV-N13 plus pCD24 group. (**G**) IL-13 response: The resu lts are comparable to those of IFN-gamma. (**H**) TNF-alpha response: pV-N13 plus pCD24 construct stimulated the highest levels of TNF-alpha. * *p* < 0.05; ***p* < 0.01, and *** *p* < 0.001 versus pVAX-1 group. All of the data are shown as mean ± SD.

**Figure 6 viruses-11-00075-f006:**
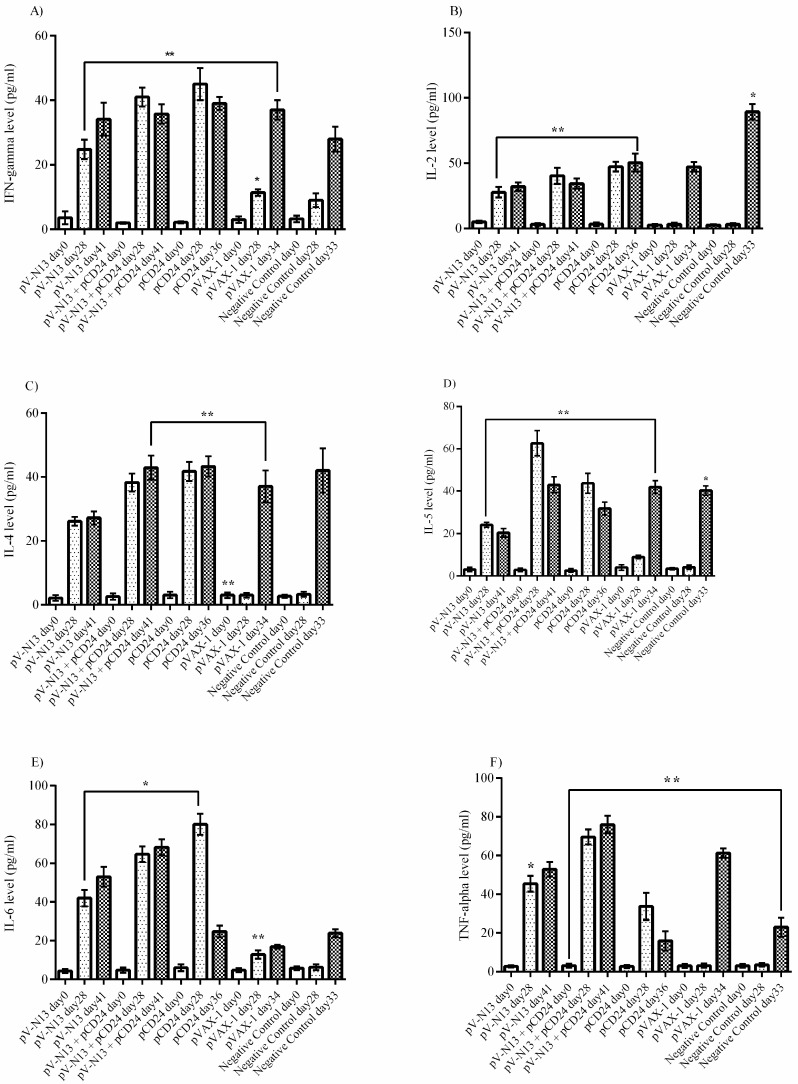
Cytokine responses in the serum samples of the immunized IFNAR^−/−^ mice. (**A**) IFN-gamma response: pV-N13 and pV-N13 plus pCD24 showed relatively high levels of IFN-gamma on day 28 (before challenge). Despite pV-N13 plus pCD24, in the pV-N13 group, this cytokine was elevated on day 41 (13 days after challenge). (**B**) IL-2 response: The results of IL-2 are comparable to INF-gamma. (**C**) IL-4 response: pV-N13 and pV-N13 plus pCD24 demonstrated the potential to stimulate IL-4 before challenge on day 28 and 13 days after challenge on day 41. (**D**) IL-5 response: pV-N13 plus pCD24 is predominant compared with pV-N13. However, both groups demonstrated a decrease on day 41. (**E**) IL-6 response: pV-N13 plus pCD24 appeared more immunogenic than pV-N13 via the IL-6 levels. (**F**) TNF-alpha response: The results are comparable to IL-6 and both N expressing constructs had the potential to elicit adequate TNF-alpha responses before and after challenge. * *p* < 0.05; ** *p* < 0.01, and *** *p* < 0.001 versus pVAX-1 group. All of the data are shown as mean ± SD.

**Figure 7 viruses-11-00075-f007:**
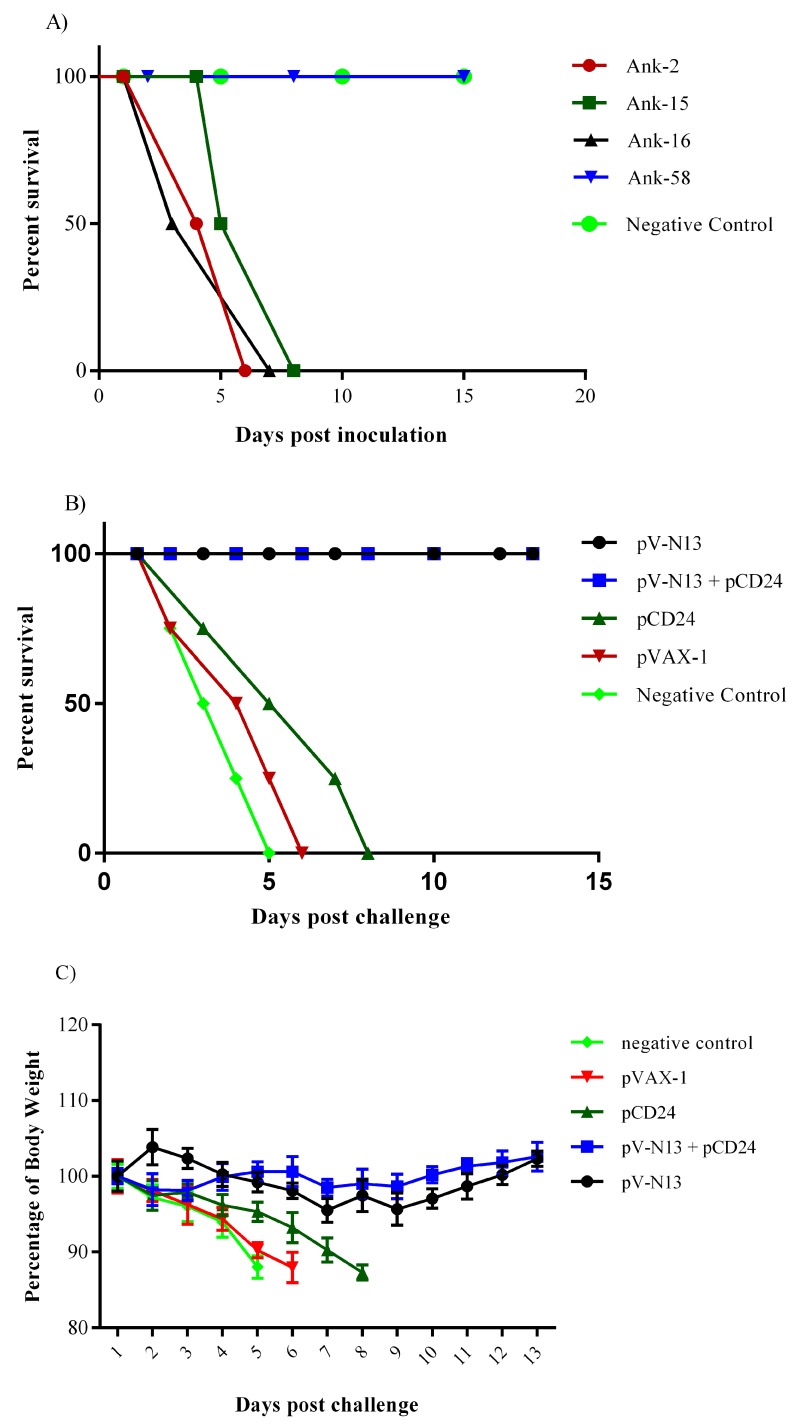
Challenge experiment. (**A**) Challenge assay to find the suitable strain of CCHFV: Four different isolates were assayed in IFNAR^−/−^ to identify the lethal strains. (**B**) Survival rate in the challenge assay with Ank-2 strain: the pV-N13 and pV-N13 plus pCD24 groups survived in the lethal dose challenge of the IFNAR^−/−^ mice. (**C**) Percentage of body weight: Despite the lethal challenge, the pV-N13 and pV-N13 plus pCD24 groups showed an almost stable body weight range. All of the data are shown as mean ± SD.

**Figure 8 viruses-11-00075-f008:**
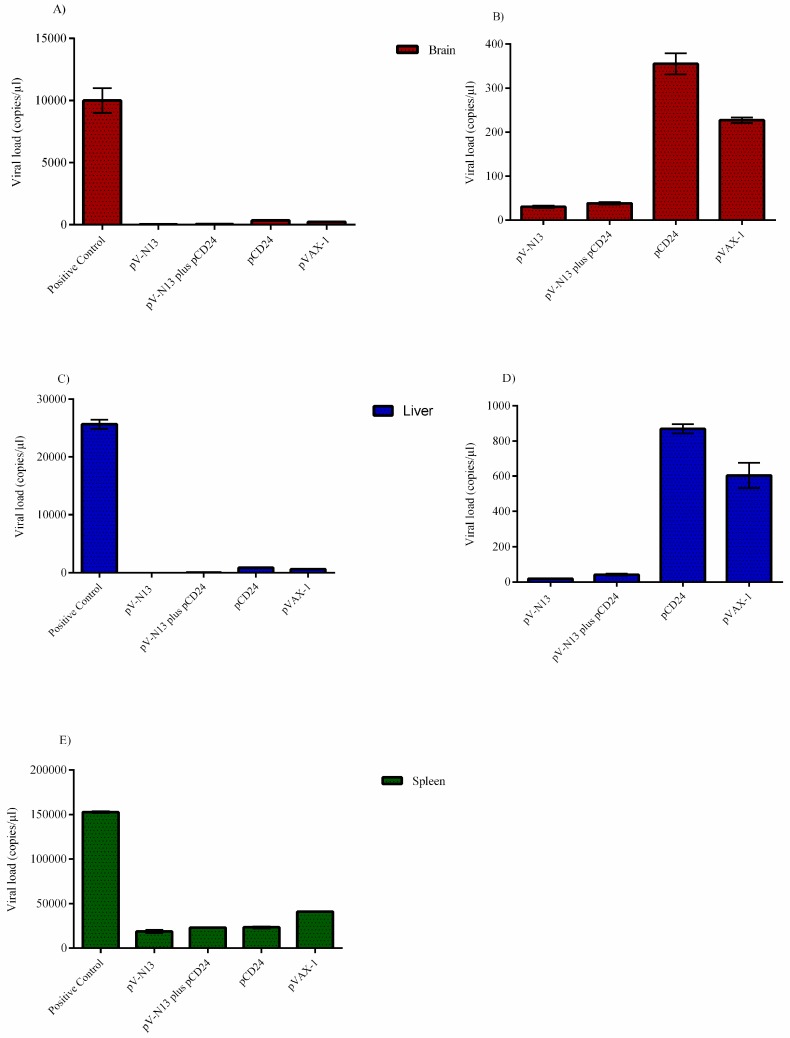
Viral loads (copy/µL of 100 ng RNA) in the tissues of the challenged IFNAR^−/−^ mice infected with 1000 TCID_50_ of the Ank-2 strain. The virus copy numbers in the brain (A: all groups; B: all groups except the positive control), liver (C: all groups; D: all groups except the positive control), and spleen (E: all groups) are provided. The most significant virus clearance was observed in the pV-N13 and pV-N13 plus pCD24 groups. All of data are shown as mean ± SD.

**Table 1 viruses-11-00075-t001:** Primer sequences employed for Crimean Congo hemorrhagic fever virus (CCHFV) N amplification and plasmid construction. Underlined bold letters indicate up- and down-stream homologous sequences of the plasmid multiple cloning site (EcoRI) of the pVAX-1 vector used in the Seamless Ligation Cloning Extract (SLiCE) cloning method.

Name	Sequence (5′-3′)
CCHFV-N-F ^a^	atggaaaacaagatcgagg
CCHFV-N-R ^a^	aggaggagaaaagctgaa
pVAX-SliCE-N-F ^b^	**taagcttggtaccgagctcggatccactagtccagtgtggtggacc**atggaaaacaagatcgagg
pVAX-SliCE-N-R ^b^	**actcgagcggccgccactgtgctggatatctgcagaatt**aggaggagaaaagctgaa
CD24-F ^c^	acccacgcagatttattcca
CD24-R ^c^	accacgaagagactggctgt
RE-N-F ^d^	gaattcatggaaaacaagatcgagg
RE-N-R ^d^	ctcgagaggaggagaaaagctgaa
qPCR-F	ggacataggtttccgtgtca
qPCR-R	tccttctaatcatgtctgacagc
qPCR-probe	FAM-agaacaacttgccaattaccaacaggc-BHQ1

^a^ Based on the CCHFV Turkey-Kelkit06 complete sequence (GenBank accession: GQ337053). ^b^ Underlined bold letters indicate homologous recombination sequences.: ^c^ Based on the CD24-expressing Open Reading Frame (ORF) (GenBank accession: NM013230). ^d^ Underlined regular letters indicate restriction endonuclease recognition sites (EcoRI and XhoI).

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
