# Peer review of "Co-Delivery Effect of CD24 on the Immunogenicity and Lethal Challenge Protection of a DNA Vector Expressing Nucleocapsid Protein of Crimean Congo Hemorrhagic Fever Virus"

_viruses, 2019, doi:10.3390/v11010075_

Reviewer 1 Report

In the manuscript “Adjuvant Potential of CD24 on Immunogenicity and Lethal Challenge Protection of a DNA vector Expressing Nucleocapsid protein of a Crimean Congo Hemorrhagic Fever Virus”, the authors describe a DNA-based vaccinal approach against Crimean Congo Hemorrhagic Fever Virus (CCHFV). They propose that, when expressed together with the viral nucleocapsid, the protein CD24 acts as a prominent adjuvant to protect mice against CCHFV challenge. This study intends to show that mice immunized with viral nucleocapsid protein-expressing plasmid can confer protection against lethal CCHFV challenges. This first part is not new and has already been documented in the literature. The real novelty here is that CD24 might be used as an adjuvant to elevate the overall level of immune responses, which might help to protect against pathogen invasion. However, that there is no difference in protection against CCHFV in IFNAR-/- mice in the presence or not of CD24 addresses the question about the relevance to use CD24 as an adjuvant in that case. At the end, the advantage of using CD24 under natural conditions remains highly hypothetical regarding CCHFV. Immune response should have been investigated in IFNAR-/- mice instead of BALB/c, with the goal to identify the molecular/immunological link(s) with an eventual increase in the protection mediated by CD24 itself. I could not find information regarding the way that authors immunized the mice with DNAs in the text (only vaguely mentioned in the Figure 1). It is critical to provide detailed information on the methods to the reader or to make it easier to find and more apparent. The figures must overall be improved. Resolution is rather low, and many are difficult to assess. I have also other major concerns that should be addressed by the authors (see below).

Specific points

1. Both UK and US English are used and authors should remain with only one of the two.

2. Some abbreviations are not introduced when used for the first time, e.g. VAN in the abstract.

3. Legends for Figures 4 and 5 are incorrect and the same mistakes are also in the results.

4. Figures 2A and 2B should have the same brightness settings, 2B is very dark compared to 2A.

5. In the text the authors write that Figure 3F shows that the ratio IgG2a/IgG1 exceeds 1 in the pCD24 groups, but all the ratios are around 0.5, with the exception of the saline control.

6. Authors say that the serum was used to determine the virus neutralizing efficiency, but data are absent and must be shown.

7. In Figure 3G-J negative controls are missing. Cells not expressing N should be used to assess the background level, and serum from all groups should also be shown.

8. Throughout the manuscript (Figures 3 to 5) the saline group has been used to define the background, but pVAX-1 group should be used instead.

9. Figures 4 and 5 are nice looking and easy to read. The same layout should be employed for the other figures.

10. One thing lacking in Figure 4 is splenocytes. They were not challenged with CCHFV. What are the levels of the cytokines from these cells? Further, are these cells infected and if yes how many? CCHFV infection cause cell death in some cell types, what is the survival ratio for these cells after CCHFV challenge?

10. The * in the Figures 4 and 5 should be shown in a way that is easier to follow. As it is currently shown, there can be misinterpretations.

11. As IFNAR-/- mice are the main model to study CCHFV challenge, why were the cytokine levels in these cells not studied? Although these cells are interferon deficient, they will still elicit cytokine responses.

12. In Figure 6B, there is no data for survival from 11 to 13 days after CCHFV challenging. Figure 6 clearly demonstrates that the N protein might be used as vaccine candidate but the role of CD24 is vague at the best. Th1 and Th2 responses are discussed based on the cytokine data but these cells are not studied, which would represent a significant gain for the story.

Author Response

First of all, all the authors really appreciate your precise and insightful review. All the mentioned points are correct and helpful. We addressed all the issues here in red by mentioning the line number. Thanks for your time.

CD24 expression plasmid clearly stimulates immune responses in the BALB/c and IFNAR-/- mice. Since single administration of pV-N13 also appears as protective in challenged IFNAR-/- mice, it is hard to evaluate the adjuvant potential of CD24 when co-injected with the N expressing construct. However, a milder disease is observed in CD24 plus pV-N13 group, based on assessments of clinical signs and body weight after virus challenge, which indirectly indicates an obvious beneficial effect of CD24 co-expression. We agree that the impact of CD24 as an adjuvant cannot be confirmed via analyzing cytokine responses. Therefore, we modified the manuscript title and the outcome of CD24 is described as a “synergistic effect” throughout the manuscript. During revision, we have further provided detailed descriptions of some of the relevant methods/experiments as well as figures with higher resolution.

Specific Points

1. Both UK and US English are used and authors should remain with only one of the two.

The authors, supported by a native English speaker, have checked and edited several linguistic errors in the text. However, we are open to use MDPI English editing system if regarded as necessary.

2. Some abbreviations are not introduced when used for the first time, e.g. VAN in the abstract.

We have checked and corrected the abbreviations in the revised manuscript.

3. Legends for Figures 4 and 5 are incorrect and the same mistakes are also in the results.

We corrected the legends and order of appearance in the text.

4. Figures 2A and 2B should have the same brightness settings, 2B is very dark compared to 2A.

The mentioned figures replaced with new ones.

5. In the text the authors write that Figure 3F shows that the ratio IgG2a/IgG1 exceeds 1 in the pCD24 groups, but all the ratios are around 0.5, with the exception of the saline control.

Actually, we tried to show an overall shift towards Th1 or Th2 responses. We corrected the manuscript as requested.

6. Authors say that the serum was used to determine the virus neutralizing efficiency, but data are absent and must be shown.

Line 290: It has been mentioned in the results that this assay was negative for all the groups.

7. In Figure 3G-J negative controls are missing. Cells not expressing N should be used to assess the background level, and serum from all groups should also be shown.

To check the background, we included serum samples from other immunization groups including empty vector, CD24 expressing and normal saline. As observed, we had some spontaneous positive cells in the control groups which were not significant and had no effect on the results of the study groups.

8. Throughout the manuscript (Figures 3 to 5) the saline group has been used to define the background, but pVAX-1 group should be used instead.

We modified the analyses and used empty vector as background. This affected p values of certain groups, which are provided in the revised figures.

9. Figures 4 and 5 are nice looking and easy to read. The same layout should be employed for the other figures.

As pointed out by the referee, all figures checked and edited as indicated.

10. One thing lacking in Figure 4 is splenocytes. They were not challenged with CCHFV. What are the levels of the cytokines from these cells? Further, are these cells infected and if yes how many? CCHFV infection cause cell death in some cell types, what is the survival ratio for these cells after CCHFV challenge?

Actually, we analyzed the naive and virus stimulated splenocytes from normal saline groups and added it here. CCHFV caused considerable toxicity in the splenocytes of negative control mice.

10. The * in the Figures 4 and 5 should be shown in a way that is easier to follow. As it is currently shown, there can be misinterpretations.

We have made the resolution of the photos more.

11. As IFNAR-/- mice are the main model to study CCHFV challenge, why were the cytokine levels in these cells not studied? Although these cells are interferon deficient, they will still elicit cytokine responses.

We fully agree and have performed cytokine analysis in IFNAR-/- mice sera. The data have been included in the revised manuscript.

12. In Figure 6B, there is no data for survival from 11 to 13 days after CCHFV challenging.

Figure 6 clearly demonstrates that the N protein might be used as vaccine candidate but the role of CD24 is vague at the best. Th1 and Th2 responses are discussed based on the cytokine data but these cells are not studied, which would represent a significant gain for the story.

 Reviewer 2 Report

In “Adjuvant Potential of CD24 on Immunogenicity and Lethal Challenge Protection of a DNA vector Expressing Nucleocapsid Protein of Crimean Congo Hemorrhagic Fever Virus” Farzani et al. describe the development of a DNA vaccine platform expressing N, and determined its effectiveness alone, or in combination with a potential adjuvant, CD24. In both cases, the platform demonstrated 100% protection using a prime boost inoculation strategy. The paper is informative, although the data is primarily descriptive with minimal effort taken to determine specifics of immunity beyond describing antibody and cytokine levels. While well organized, there are sections that would benefit from additional editing, and there are several important details in the methods that are absent.

 General comments:

-As mentioned in the discussion, the adjuvant potential of CD24 in conjunction with pV-N13 cannot be reported on because both products protected when given alone, although a decrease in disease severity was clear. Only one combination of pV-N13+CD24 was described. One solution to this would be to use different combination of the two platforms at increasing lower concentration until breakthrough was observed. At these levels it should possible to interpret the various protection levels and potential synergy of both platforms. Based on this issue, I would recommend reconsidering the title of the report.

-Report would be strengthened by developing on the in vivo work with the immunized IFNAR knockout mice. Figure 6a describes virulence in this model for four Turkish strains of CCHFV. Data relating to virulence in strains outside of the standard IbAr10200 challenge model are scarce, expanding this work to report weight loss, clinical signs, sequence data, etc., would be of benefit to the field. A heterologous challenge using other CCHFV strains from more geographically diverse clades would also greatly enhance the impact of this report.

-Nothing is reported on the status of the mice (both BALB/c and IFNAR KO) during the vaccination periods. Even if uneventful, a sentence reporting the presence or absence of any clinical signs should be included.

 Specific Comments

-Line 44: Please reword. As written, “prior to human infection” suggests that studies referenced involved tick passage followed by experimental infection in humans.

-Line 79: include details of criteria for euthanasia. Weight loss of certain %? Specific clinical signs? Point scale?

-Line 93: please include more details on CCHFV ANK-2, including original isolation information (source? Geographical location? Reference?), passage history, sequence (submitted to genbank?)

-Line 146: describe the selective antibiotic regime – was it the same as the initial selection (if so state) or different?

-Line 149: specify gender of mice and source

-Line 150: What was the site of IM inoculation

-Line 151: rationale for tail vein blood collection?

-Fig 1: Why was 40 ug of pV-N13 and 10 ug of pCD24 plasmid chosen for the combination group? Were other combinations attempted?

-Line 157: details these viral strains are absent and should be included (genbank numbers, phylogenetics, clades, source of isolation. If human isolates, please state disposition of patients these strains were isolated from)

-Line 158: what volume of virus was injected into the mice?

-Line 162: define what 100 LD50 equates to in PFU or TCID50

-Fig 3: Hard to interpret G-J without seeing the monolayer in BF, especially without a control panel.

-Fig 4: Stats look off for some of these (especially B and D)

-Figures 4 and 5 could be combined to include only the most relevant data, with the rest moved to a separate supplementary figure.

-Fig 6: Is too small to read the legend, etc.

-Fig 6A: What was different about the strain that caused no disease? Did you do PCR on tissues/look at ab titers in serum to confirm infection?

-Fig 6B: why does data in 6B and 6C end on different days? Day 10 and 13 post-challenge, respectively

-Fig 6C: Please clarify what is depicted in 6c -- Is it the mean? SD or range should also be shown, at a minimum.

-Line 313: references needed for statement re: IFNAR and STAT1 mice

Author Response

First of all, all the authors really appreciate your precise and insightful review. All the mentioned points are correct and helpful. We addressed all the issues here in red by mentioning the line number. Thanks for your time.

General comments:

-As mentioned in the discussion, the adjuvant potential of CD24 in conjunction with pV-N13 cannot be reported on because both products protected when given alone, although a decrease in disease severity was clear. Only one combination of pV-N13+CD24 was described. One solution to this would be to use different combination of the two platforms at increasing lower concentration until breakthrough was observed. At these levels it should possible to interpret the various protection levels and potential synergy of both platforms. Based on this issue, I would recommend reconsidering the title of the report.

In the present study, we have shown that CD24 when expressed alone has the potential to elevate the immune responses in the BALB/c and IFNAR-/- mice. However, because pV-N13 alone is protective in challenge IFNAR-/- mice, it is hard to evaluate the adjuvant potential of CD24 when co-injected with our N expressing construct. Despite this fact, we have to mention that in CD24 plus pV-N13 group, the disease severity is decreased, based on assessments of clinical signs and stability of body weight after challenge with lethal dose of CCHFV. However, it is impossible to verify CD24 as an adjuvant by analyzing cytokine responses. Therefore, we omitted the word adjuvant in the text and called the outcome as a “synergistic effect”, which better describes our story. Also, we changed the title of the manuscript. On the other hand, we increased the resolution of the figures to 600 bpi and the immunization schedule completely described in the method section.

-Report would be strengthened by developing on the in vivo work with the immunized IFNAR knockout mice. Figure 6a describes virulence in this model for four Turkish strains of CCHFV. Data relating to virulence in strains outside of the standard IbAr10200 challenge model are scarce, expanding this work to report weight loss, clinical signs, sequence data, etc., would be of benefit to the field. A heterologous challenge using other CCHFV strains from more geographically diverse clades would also greatly enhance the impact of this report.

We have collected the serum samples from challenged IFNAR-/- mice and performed cytokine analysis and added these results to the manuscript. This research was a part of a larger project to develop an efficient regional vaccine against the virus. So, we decided to use the local isolates recovered from infected humans in Turkey. As you mentioned, virulence of local strains is not completely understood. So, we used 4 different isolates and as you know, one of them was unable to cause death in the IFNAR-/- mice.

-Nothing is reported on the status of the mice (both BALB/c and IFNAR KO) during the vaccination periods. Even if uneventful, a sentence reporting the presence or absence of any clinical signs should be included.

We included the clinical symptoms in the revised manuscript. In BALB/c, we only conducted vaccine immunization and observed no signs including allergic reactions to our constructs. In the case of IFNAR-/-, we included clinical signs in the main text.

Specific Comments

-Line 44: Please reword. As written, “prior to human infection” suggests that studies referenced involved tick passage followed by experimental infection in humans.

-Line 45: We have rephrased the statement

-Line 79: include details of criteria for euthanasia. Weight loss of certain %? Specific clinical signs? Point scale?

Line 80: Criteria of euthanasia were added in the method section.

-Line 93: please include more details on CCHFV ANK-2, including original isolation information (source? Geographical location? Reference?), passage history, sequence (submitted to genbank?)

Line 100: This strain was isolated from human blood sample and has been submitted to GenBank. We provided the access number of this strain and Ank-58 in the revised manuscript. All the requested data including source, geographical location, reference, passage history and sequence are added to the method section.

-Line 146: describe the selective antibiotic regime – was it the same as the initial selection (if so state) or different?

Line 162: We clarified the selection regime in the main text.

-Line 149: specify gender of mice and source

Line 85 and 167: It was added in the methods section.

-Line 150: What was the site of IM inoculation

Line 172: The inoculation was performed using the thigh muscles. Added to the revised text. 

-Line 151: rationale for tail vein blood collection?

Line 173: It is a routine protocol in the department to collect the blood from the tail due to its simplicity.

-Fig 1: Why was 40 ug of pV-N13 and 10 ug of pCD24 plasmid chosen for the combination group? Were other combinations attempted?

Line 169: We have previously tried different combinations of pV-N13 to pV-N13 plus pCD24 (1:1 to 5:1) and the best results obtained in 4:1 based on antibody and cytokine responses in BALB/c mice.

-Line 157: details these viral strains are absent and should be included (genbank numbers, phylogenetics, clades, source of isolation. If human isolates, please state disposition of patients these strains were isolated from)

Line 181: All the strains were isolated from human blood samples from Ankara, Turkey and mentioned in the manuscript.

-Line 158: what volume of virus was injected into the mice?

Line 185: 300 microliter per mice (added to the mentioned section)

-Line 162: define what 100 LD50 equates to in PFU or TCID50

Line 189: This was added to the mentioned section

-Fig 3: Hard to interpret G-J without seeing the monolayer in BF, especially without a control panel.

Fig 3: We added the serum results from the other groups as background. Some spontaneous positive cells can be seen in the control groups which are not significant in comparison to the main groups (N expressing ones).

-Fig 4: Stats look off for some of these (especially B and D)

Fig 4: We changed the resolution of the figures and increased to 600 bpi.

-Figures 4 and 5 could be combined to include only the most relevant data, with the rest moved to a separate supplementary figure.

-Fig 6: Is too small to read the legend, etc.

 Figure 6 was redrawn.

-Fig 6A: What was different about the strain that caused no disease? Did you do PCR on tissues/look at ab titers in serum to confirm infection?

-Fig 6A: In this case, we conducted TCID50 assay from organs and verified the virus replication in the mice. In addition, sequencing of the S segment revealed 99% identity with the local isolates. It is possible that the difference may have arisen due to variations in other segments

-Fig 6B: why does data in 6B and 6C end on different days? Day 10 and 13 post-challenge, respectively

-Fig 6B: The mentioned figure was corrected.

-Fig 6C: Please clarify what is depicted in 6c -- Is it the mean? SD or range should also be shown, at a minimum.

-Fig 6C: The results are provided as mean.

-Line 313: references needed for statement re: IFNAR and STAT1 mice

Line 417: We added the reference.

 Round  2

Reviewer 1 Report

Authors carefully answered all my points and I do understand the limitations encountered by authors in the study. However, I still consider that the title is inappropriate according to the data shown. Synergic effect with CD24 is only seen at the cytokine levels as no difference is observed in the survival of mice. This suggests that the level of cytokines in animals is sufficient to protect mice even in the absence of CD24. One could argue in the discussion that CD24 offers a stronger protection but it remains at the level of discussion since no data supporting this is shown. Consequently, in the absence of further data, I strongly suggest, at least, to rephrase the title to match closer to the results.

Author Response

Dear Professor

We deeply appreciate your comments. Your mentioned issues greatly improved our manuscript. As you noted, CD24 effect can only be seen in the cytokine level, so it is not possible for us to tell that it has an adjuvant or synergistic effect alongside with N expressing construct. So we changed the title to (Co-delivery effect of CD24) and added some sentences (in red) in the discussion to clarify this fact. 

Reviewer 2 Report

The authors have made a notable effort to address the majority of my comments and the revised manuscript is greatly improved. I have just one more minor comment.

 Fig.7 is shown as mean only, please include error bars or show as individual data with a mean.

Author Response

Dear Professor

We deeply appreciate your valuable comments on our manuscript which lead to an improved version. We corrected the Figure 7 and changed the data into Mean+SD, also included error bar in the graph.